# QUANTUM KOLMOGOROV-ARNOLD NETWORKS

## ABSTRACT

The pursuit of quantum advantage in machine learning drives the exploration of quantum analogues of powerful classical architectures. The recent introduction of Kolmogorov-Arnold Networks (KANs) provides a mathematically grounded framework for enhanced expressivity and accuracy in function approximation. However, KANs and their variants fundamentally rely on predefined basis functions. In this work, we introduce Quantum Kolmogorov-Arnold Networks (QKANs), which leverage parameterized quantum circuits to implement learnable activation functions without predefined bases. We establish the theoretical foundations of QKANs and demonstrate their effectiveness through numerical experiments, showing superior performance on function approximation tasks. QKANs accurately model complex nonlinear relationships and establish a new benchmark for expressive power in quantum machine learning. This work bridges KANs with quantum computation, providing a new paradigm for expressive quantum machine learning.

## 1 INTRODUCTION

Efficient function approximation is a cornerstone of machine learning, with neural networks serving as powerful tools for modeling complex nonlinear relationships. A foundational result in this domain is the Kolmogorov–Arnold representation theorem, originally developed to address Hilbert's 13th problem (Girosi & Poggio, 1989). Recent advances have revitalized this theorem through Kolmogorov–Arnold Networks (KANs) (Liu et al., 2025), which combine insights from Kolmogorov networks (Schmidhuber, 1997; Sprecher & Draghici, 2002) with learnable activation functions (Fakhoury et al., 2022). KANs demonstrate impressive accuracy, efficiency, and interpretability across mathematical and physical science problems (Liu et al., 2025), positioning them as compelling alternatives to Multi-Layer Perceptrons (MLPs).

The introduction of KANs has rapidly inspired diverse architectural innovations across multiple domains. For graph neural networks, Kolmogorov–Arnold Attention (KAA) (Fang et al., 2025) integrates KANs into attention mechanisms. The Kolmogorov–Arnold Transformer (KAT) (Xingyi Yang, 2025) replaces standard MLP blocks with KAN layers to boost transformer expressiveness. Physics-Informed Kolmogorov-Arnold Networks (PIKANs) extended KANs to physics-informed learning for modeling complex system dynamics (Shuai & Li, 2025). KANs have also demonstrated strong performance in Out-of-Distribution (OOD) scenarios by exploiting their inherent local plasticity for improved generalization (Canevaro et al., 2025). Applications further extend to smart grid security (Wu et al., 2025), battery storage systems (Zequera et al., 2025), genomic analysis (Cherednichenko & Poptsova, 2025), time series forecasting (Huang et al., 2025), quantum architecture search (Kundu et al., 2024) and beyond (Del Rosario et al., 2024; Seydi et al., 2025; Zhong et al., 2024).

In parallel, quantum machine learning seeks to harness quantum mechanical principles to enhance classical learning paradigms (Biamonte et al., 2017; Jerbi et al., 2023). With the advent of the Noisy Intermediate-Scale Quantum (NISQ) era, research emphasis shifted toward Variational Quantum Circuits (VQCs) (Abbas et al., 2021; Cerezo et al., 2021; Manzano et al., 2025; Yu et al., 2024)—trainable quantum networks optimized using classical gradient-based methods. A central theme in quantum machine learning is the development of quantum analogues of successful classical models, such as Quantum Convolutional Neural Networks (QCNNs) (Cong et al., 2019) and Quantum Recurrent Neural Networks (QRNNs) (Bausch, 2020).

This work is motivated by a notable structural alignment: the inherent compatibility between KANs and quantum computation. We introduce Quantum Kolmogorov–Arnold Networks (QKANs), providing a detailed mathematical formulation and corresponding quantum circuit designs. Since quantum circuits naturally implement polynomial transformations through parameterized gates and established subroutines (Motlagh & Wiebe, 2024), QKANs can leverage quantum properties to construct learnable activation functions without relying on predefined classical basis functions.

## 2 RELATED WORK

A central theme in the development of KANs has been the search for effective basis functions to parameterize their learnable activation functions. The foundational work used B-splines, which has since inspired a family of variants that substitute different basis functions, such as Fourier series (Guo et al., 2025; Pang et al., 2025), Chebyshev polynomials (Mahmoud et al., 2025), and Legendre polynomials (Chen et al., 2025).

In parallel, researchers have begun to adapt KANs for execution on quantum hardware. Ivashkov et al. (2024) pioneered a quantum KAN that uses block encoding and the Quantum Singular Value Transformation (QSVT) (Gilyén et al., 2019) to generate Chebyshev polynomials, requiring a weighted sum to approximate activation functions. Our work departs from this approach in several fundamental ways. While their architecture operates within the standard block encoding paradigm, ours is constructed using the more general framework of (Hermitian) projected unitary encodings (Gilyén et al., 2019), offering greater versatility for circuit construction. The core technical distinction lies in polynomial synthesis: their method uses QSVT to generate Chebyshev polynomials, requiring a weighted sum to approximate arbitrary activations, whereas our approach leverages Generalized Quantum Signal Processing (GQSP) (Motlagh & Wiebe, 2024) to natively and directly implement arbitrary polynomials, including those of indefinite parity, without confinement to a fixed basis. A further novel contribution is our explicit procedure for completing general encodings to Hermitian ones, a critical step to ensure consistency across network layers that their framework does not address. Finally, beyond theoretical construction, we provide empirical validation through numerical simulations, demonstrating the feasibility of our basis-free architecture, while their work remains purely analytical.

Similarly, Werner et al. (2025) developed a variant that embeds data amplitudes via pre-trained Quantum Circuit Born Machines (QCBMs) (Du et al., 2020) to represent classical predetermined basis functions, using parameterized entangling layers to optimize the coefficients in a linear combination. Although data transformation occurs on the quantum device, this method still relies on a predetermined basis set.

These quantum adaptations inherit a core constraint from their classical counterparts: a dependence on the basis function paradigm, where learning an activation function reduces to fitting coefficients within a fixed, pre-selected basis. Our work challenges this fundamental paradigm. We propose that the native expressivity of parameterized quantum circuits is itself sufficient for learning activation functions directly. By leveraging the GQSP—which natively implements arbitrary polynomials without an intermediate basis—and by encoding a single scalar per subroutine, we eliminate the need for basis selection entirely. This enables the implementation of a fully learnable, basis-free activation function directly on the quantum processor.

## 3 METHOD

### 3.1 ENCODING

Our method leverages the framework of (Hermitian) projected unitary encoding (Gilyén et al., 2019) to embed classical data into quantum states.

**Definition 1** (Projected unitary encoding). *Let $A \in \mathbb{C}^{N_L \times N_R}$ satisfy $\|A\|_2 \leq 1$. Given a unitary matrix $U \in \mathbb{C}^{M \times M}$ and isometries $\Pi_L \in \mathbb{C}^{M \times N_L}$ and $\Pi_R \in \mathbb{C}^{M \times N_R}$, the tuple $(U, \Pi_L, \Pi_R)$ is called a projected unitary encoding of $A$ if*

$$\Pi_L^\dagger U \Pi_R = A. \tag{1}$$

**Definition 2** (Hermitian projected unitary encoding). *Let $A \in \mathbb{C}^{N \times N}$ be Hermitian. Given a unitary matrix $U \in \mathbb{C}^{M \times M}$ and an isometry $\Pi \in \mathbb{C}^{M \times N}$, the tuple $(U, \Pi)$ is called a Hermitian projected unitary encoding of A if U is a Hermitian unitary matrix and $(U, \Pi, \Pi)$ forms a projected unitary encoding of A.*

We construct a projected unitary encoding for each scalar component. For any real input $x \in (-1, 1)$, we define the base encoding via the $2 \times 2$ Hermitian unitary matrix

$$U(x) := \begin{bmatrix} x & \sqrt{1-x^2} \\ \sqrt{1-x^2} & -x \end{bmatrix}. \tag{2}$$

The matrix $U(x)$ is both Hermitian and unitary. In particular, the tuple $(U(x), |0\rangle)$ constitutes a Hermitian projected unitary encoding of the scalar $x$, since

$$\langle 0|U(x)|0\rangle = x. \tag{3}$$

This subroutine can be implemented by a single-qubit gate. As the quantum circuit evolves, we recursively compose established subroutines. At each stage, we obtain a unitary $U$ corresponding to the subroutine and an isometry $|\psi\rangle$ such that

$$\Re(\langle\psi|U|\psi\rangle) = y, \tag{4}$$

where $y$ denotes the result at this stage.

## 3.2 QUANTUM POLYNOMIAL TRANSFORMATION

We establish a framework for transforming real polynomials using quantum operations, beginning with the necessary definitions, proving two key lemmas, and culminating in a theorem for encoding arbitrary real polynomials into unitary circuits.

**Definition 3** (Generalized single-qubit rotation operator). *For real parameters $\theta, \phi, \lambda \in \mathbb{R}$, the generalized single-qubit rotation operator is defined as*

$$R(\theta, \phi, \lambda) := \begin{bmatrix} e^{i(\lambda+\phi)}\cos\theta & e^{i\phi}\sin\theta \\ e^{i\lambda}\sin\theta & -\cos\theta \end{bmatrix}. \tag{5}$$

**Definition 4** (Generalized quantum signal processing operator). *Let U be a unitary operator, $d \in \mathbb{N}$, and let $\boldsymbol{\theta} := (\theta_0, \ldots, \theta_d) \in \mathbb{R}^{d+1}$, $\boldsymbol{\phi} := (\phi_0, \ldots, \phi_d) \in \mathbb{R}^{d+1}$, and $\lambda \in \mathbb{R}$. The generalized quantum signal processing operator is defined as*

$$\hat{Q}(U; \boldsymbol{\theta}, \boldsymbol{\phi}, \lambda) = \left(\prod_{j=1}^{d} (R(\theta_j, \phi_j, 0) \otimes I)(|0\rangle\langle 0| \otimes U + |1\rangle\langle 1| \otimes I)\right)(R(\theta_0, \phi_0, \lambda) \otimes I), \tag{6}$$

**Definition 5** (Reflection operator). *Let $\mathcal{H}$ be a Hilbert space, and let $|\psi\rangle \in \mathcal{H}$ be a unit vector. The reflection operator $\mathcal{R}_\psi$ about the state $|\psi\rangle$ is defined as*

$$\mathcal{R}_\psi := 2|\psi\rangle\langle\psi| - I. \tag{7}$$

**Lemma 6.** *Let $(U, |\psi\rangle)$ be a Hermitian projected unitary encoding of the real scalar $x \in (-1, 1)$. Define*

$$|\psi^\perp\rangle := \frac{1}{\sqrt{1-x^2}}(U|\psi\rangle - x|\psi\rangle). \tag{8}$$

*Then $|\psi^\perp\rangle$ is a unit vector orthogonal to $|\psi\rangle$, and the two-dimensional subspace $\mathcal{S} = \text{span}\{|\psi\rangle, |\psi^\perp\rangle\}$ is invariant under U. Moreover, with respect to the basis $\{|\psi\rangle, |\psi^\perp\rangle\}$, the restriction of U to $\mathcal{S}$ is*

$$U|_{\mathcal{S}} = \begin{pmatrix} x & \sqrt{1-x^2} \\ \sqrt{1-x^2} & -x \end{pmatrix}. \tag{9}$$

*Proof.* We first verify that $|\psi^\perp\rangle$ is normalized:

$$
\begin{aligned}
\left\| |\psi^\perp\rangle \right\|^2 &= \left\| \frac{1}{\sqrt{1-x^2}} (U|\psi\rangle - x|\psi\rangle) \right\|^2 \\
&= \frac{1}{1-x^2} (U^\dagger \langle\psi| - x\langle\psi|)(U|\psi\rangle - x|\psi\rangle) \\
&= \frac{1}{1-x^2} \left( \langle\psi|U^2|\psi\rangle - 2x\langle\psi|U|\psi\rangle + x^2\langle\psi|\psi\rangle \right) \\
&= \frac{1}{1-x^2} \left( 1 - 2x^2 + x^2 \right) \\
&= \frac{1-x^2}{1-x^2} \\
&= 1.
\end{aligned}
\tag{10}
$$

To check orthogonality:

$$
\langle\psi|\psi^\perp\rangle = \frac{1}{\sqrt{1-x^2}} (\langle\psi|U|\psi\rangle - x\langle\psi|\psi\rangle) = \frac{1}{\sqrt{1-x^2}}(x-x) = 0.
\tag{11}
$$

Next, we show that $\mathcal{S}$ is invariant under $U$ by expressing $U|\psi\rangle$ in the basis $\{|\psi\rangle, |\psi^\perp\rangle\}$:

$$
U|\psi\rangle = x|\psi\rangle + \sqrt{1-x^2}|\psi^\perp\rangle
\tag{12}
$$

Now compute $U|\psi^\perp\rangle$:

$$
\begin{aligned}
U|\psi^\perp\rangle &= \frac{1}{\sqrt{1-x^2}} \left( U^2|\psi\rangle - xU|\psi\rangle \right) \\
&= \frac{1}{\sqrt{1-x^2}} \left[ |\psi\rangle - x\left( x|\psi\rangle + \sqrt{1-x^2}|\psi^\perp\rangle \right) \right] \\
&= \sqrt{1-x^2}|\psi\rangle - x|\psi^\perp\rangle.
\end{aligned}
\tag{13}
$$

Thus, both $U|\psi\rangle$ and $U|\psi^\perp\rangle$ lie in $\mathcal{S}$, confirming that the subspace is invariant under $U$. The matrix representation of $U$ in this basis is

$$
U|_\mathcal{S} = \begin{pmatrix} \langle\psi|U|\psi\rangle & \langle\psi|U|\psi^\perp\rangle \\ \langle\psi^\perp|U|\psi\rangle & \langle\psi^\perp|U|\psi^\perp\rangle \end{pmatrix} = \begin{pmatrix} x & \sqrt{1-x^2} \\ \sqrt{1-x^2} & -x \end{pmatrix}.
\tag{14}
$$

$\square$

**Lemma 7.** *Let $(U, |\psi\rangle)$ be a Hermitian projected unitary encoding of a real scalar $x \in (-1, 1)$. Define the qubitized walk operator*

$$
W := \mathcal{R}_\psi U.
\tag{15}
$$

*Let*

$$
|\psi^\perp\rangle := \frac{1}{\sqrt{1-x^2}} (U|\psi\rangle - x|\psi\rangle),
\tag{16}
$$

*and define the states*

$$
|\Psi_\pm\rangle := \frac{1}{\sqrt{2}} \left( |\psi\rangle \pm i|\psi^\perp\rangle \right).
\tag{17}
$$

*Then $|\Psi_+\rangle$ and $|\Psi_-\rangle$ form an orthonormal basis of eigenvectors of $W$ with corresponding eigenvalues $e^{\pm i \arccos x}$.*

*Proof.* We first verify that $|\Psi_+\rangle$ and $|\Psi_-\rangle$ are orthonormal. From Theorem 6, the vectors $|\psi\rangle$ and $|\psi^\perp\rangle$ are orthonormal, so we compute:

$$
\langle\Psi_+|\Psi_+\rangle = \frac{1}{2} \left( \langle\psi|\psi\rangle + i\langle\psi|\psi^\perp\rangle - i\langle\psi^\perp|\psi\rangle + \langle\psi^\perp|\psi^\perp\rangle \right) = 1,
\tag{18}
$$

$$
\langle\Psi_-|\Psi_-\rangle = \frac{1}{2} \left( \langle\psi|\psi\rangle - i\langle\psi|\psi^\perp\rangle + i\langle\psi^\perp|\psi\rangle + \langle\psi^\perp|\psi^\perp\rangle \right) = 1,
\tag{19}
$$

$$
\langle\Psi_+|\Psi_-\rangle = \frac{1}{2} \left( \langle\psi|\psi\rangle - i\langle\psi|\psi^\perp\rangle + i\langle\psi^\perp|\psi\rangle - \langle\psi^\perp|\psi^\perp\rangle \right) = 0.
\tag{20}
$$

Thus, $|\Psi_+\rangle$ and $|\Psi_-\rangle$ form an orthonormal set.

Next, we evaluate the action of $W = \mathcal{R}_\psi U$ on $|\Psi_\pm\rangle$. From Theorem 6, the action of $U$ on the basis $\{|\psi\rangle, |\psi^\perp\rangle\}$ is

$$
\begin{aligned}
U|\Psi_\pm\rangle &= \frac{1}{\sqrt{2}}\left(U|\psi\rangle \pm iU|\psi^\perp\rangle\right) \\
&= \frac{1}{\sqrt{2}}\left[\left(x|\psi\rangle + \sqrt{1-x^2}|\psi^\perp\rangle\right) \pm i\left(\sqrt{1-x^2}|\psi\rangle - x|\psi^\perp\rangle\right)\right] \\
&= \frac{1}{\sqrt{2}}\left[\left(x \pm i\sqrt{1-x^2}\right)|\psi\rangle + \left(\sqrt{1-x^2} \mp ix\right)|\psi^\perp\rangle\right].
\end{aligned}
\tag{21}
$$

Since $\mathcal{R}_\psi |\psi\rangle = |\psi\rangle$ and $\mathcal{R}_\psi |\psi^\perp\rangle = -|\psi^\perp\rangle$, we compute

$$
\begin{aligned}
W|\Psi_\pm\rangle &= \mathcal{R}_\psi U|\Psi_\pm\rangle \\
&= \frac{1}{\sqrt{2}}\left[\left(x \pm i\sqrt{1-x^2}\right)|\psi\rangle - \left(\sqrt{1-x^2} \mp ix\right)|\psi^\perp\rangle\right] \\
&= \frac{1}{\sqrt{2}}\left[\left(x \pm i\sqrt{1-x^2}\right)|\psi\rangle \pm i\left(x \pm i\sqrt{1-x^2}\right)|\psi^\perp\rangle\right] \\
&= \left(x \pm i\sqrt{1-x^2}\right) \cdot \frac{1}{\sqrt{2}}\left(|\psi\rangle \pm i|\psi^\perp\rangle\right) \\
&= e^{\pm i\arccos x}|\Psi_\pm\rangle.
\end{aligned}
\tag{22}
$$

Thus, $|\Psi_\pm\rangle$ are eigenvectors of $W$ with eigenvalues $e^{\pm i\arccos x}$. $\qquad\square$

**Theorem 8.** *Let $(U, |\psi\rangle)$ be a Hermitian projected unitary encoding of a real scalar $x \in (-1,1)$. Define the qubitized walk operator*

$$
W := \mathcal{R}_\psi U,
\tag{23}
$$

*and*

$$
V := \hat{Q}(W; \boldsymbol{\theta}, \boldsymbol{\phi}, \lambda),
\tag{24}
$$

*where $\boldsymbol{\theta}, \boldsymbol{\phi} \in \mathbb{R}^{d+1}$, and $\lambda \in \mathbb{R}$.*

*Then $(V, |0\rangle \otimes |\psi\rangle, |0\rangle \otimes |\psi\rangle)$ is a projected unitary encoding of a polynomial of degree at most $d$.*

*Proof.* By Theorem 7, $W$ has eigenvalues $e^{\pm i\arccos x}$, with corresponding orthonormal eigenvectors

$$
|\Psi_\pm\rangle := \frac{1}{\sqrt{2}}\left(|\psi\rangle \pm i|\psi^\perp\rangle\right),
\tag{25}
$$

From GQSP (Motlagh & Wiebe, 2024), there exists a unitary polynomial transformation $P$ satisfying $|P(e^{i\theta})| \le 1$ for all $\theta \in \mathbb{R}$, such that

$$
P(W) = \langle 0|\hat{Q}(W; \boldsymbol{\theta}, \boldsymbol{\phi}, \lambda)|0\rangle.
\tag{26}
$$

To evaluate $\langle\psi|P(W)|\psi\rangle$, expand $|\psi\rangle$ in the eigenbasis of $W$ and compute:

$$
\begin{aligned}
\langle\psi|P(W)|\psi\rangle &= \frac{1}{2}((\langle\Psi_+| - \langle\Psi_-|)P(W)(|\Psi_+\rangle - |\Psi_-\rangle) \\
&= \frac{1}{2}((\langle\Psi_+| - \langle\Psi_-|)\left[P\left(e^{i\arccos x}\right)|\Psi_+\rangle - P\left(e^{-i\arccos x}\right)|\Psi_-\rangle\right] \\
&= \frac{1}{2}\left[\langle\Psi_+|P\left(e^{i\arccos x}\right)|\Psi_+\rangle + \langle\Psi_-|P\left(e^{-i\arccos x}\right)|\Psi_-\rangle\right] \\
&= \frac{1}{2}\left[P\left(e^{i\arccos x}\right) + P\left(e^{-i\arccos x}\right)\right] \\
&= p(x),
\end{aligned}
\tag{27}
$$

where

$$
p(x) := \frac{P(e^{i\arccos x}) + P(e^{-i\arccos x})}{2}.
\tag{28}
$$

Therefore, $(V, |0\rangle \otimes |\psi\rangle, |0\rangle \otimes |\psi\rangle)$ is a projected unitary encoding of the polynomial $p(x)$, which has degree at most $d$ for $x \in (-1, 1)$, completing the proof. $\square$

By tuning the parameters $\boldsymbol{\theta}, \boldsymbol{\phi} \in \mathbb{R}^{d+1}$ and $\lambda \in \mathbb{R}$, we can realize a quantum transformation for any polynomial of degree at most $d$. Since we are only interested in the real part, the effective polynomial applied to $x$ is $\Re(p(x))$. The corresponding quantum circuit, which implements the operator $V = \hat{Q}(W; \boldsymbol{\theta}, \boldsymbol{\phi}, \lambda)$, is shown below:

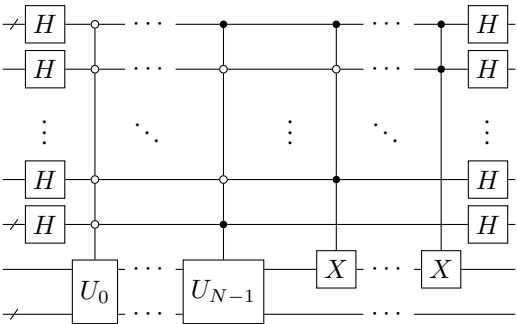

## 3.3 COMBINATION OF UNITARIES

To aggregate $N$ projected unitary encodings into a single result, we apply a normalized Linear Combination of Unitaries (LCU) (Childs & Wiebe, 2012). Given unitaries $U_i$ with isometry $|\psi\rangle$, where each $(U_i, |\psi\rangle, |\psi\rangle)$ is a projected unitary encoding of $x_i$, let $n = \lceil \log_2 N \rceil$. We define

$$V := (H^{\otimes n} \otimes I) \left( \sum_{i=0}^{N-1} |i\rangle \langle i| \otimes U_i + \sum_{i=N}^{2^n-1} |i\rangle \langle i| \otimes \tilde{U} \right) (H^{\otimes n} \otimes I), \tag{29}$$

where $\tilde{U}$ is a unitary such that $(\tilde{U}, |\psi\rangle, |\psi\rangle)$ is a projected unitary encoding of 0. This yields

$$y := \frac{1}{\sqrt{2^n}} \sum_{i=0}^{N-1} x_i, \tag{30}$$

so that $(V, |0\rangle^{\otimes n} \otimes |\psi\rangle, |0\rangle^{\otimes n} \otimes |\psi\rangle)$ forms a projected unitary encoding of $y$. In practice, we often choose $|\psi\rangle = |0\rangle^{\otimes a}$ and set $\tilde{U} = X \otimes I$.

The residual terms in the sum, $\sum_{i=N}^{2^n-1} |i\rangle \langle i| \otimes \tilde{U}$, can be implemented using $\mathcal{O}(n)$ queries to multi-controlled $\tilde{U}$. The circuit representation is shown below:

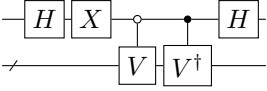

## 3.4 HERMITIAN OPERATOR COMPLETION

To ensure compatibility across layers, we complete possibly general projected unitary encoding to Hermitian projected unitary encoding. Given $(V, |\psi\rangle, |\psi\rangle)$ is a projected unitary encoding of $x \in \mathbb{C}$, we define

$$\bar{V} := (H \otimes I)(|0\rangle \langle 0| \otimes V + |1\rangle \langle 1| \otimes V^\dagger)(XH \otimes I). \tag{31}$$

This construction ensures that $(\bar{V}, |0\rangle \otimes |\psi\rangle)$ is a Hermitian encoding of $\Re(x)$. The corresponding circuit is depicted below:

### 3.5 Measurement Protocol

To extract scalar values from the output layer, we apply a Hadamard test (Cleve et al., 1998) to each final unitary $U$. This procedure estimates the real part of the matrix element $\Re(\langle\psi|U|\psi\rangle)$ by measuring the expectation value $\langle Z \rangle$ of the top qubit in the following circuit:

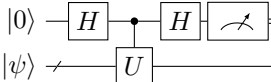

On a quantum device, classical sampling of this circuit requires $\mathcal{O}(1/\epsilon^2)$ queries to estimate $\Re(\langle\psi|U|\psi\rangle)$ within additive error $\epsilon$. This complexity can be improved to $\mathcal{O}(1/\epsilon)$ using amplitude estimation techniques (Brassard et al., 2000).

## 4 Experiments

### 4.1 Experimental Setup

To evaluate the effectiveness of the proposed method, we construct a two-layer QKAN with a hidden dimension of 5 to model functions of two variables. The model is trained with the Adam optimizer (Kingma & Ba, 2015) using a batch size of 64 and an initial learning rate of 0.25, which follows a cosine decay schedule (Loshchilov & Hutter, 2017). We perform all experiments via classical simulation and report the average over 10 independent runs with different random seeds. Input data and target outputs are scaled to a range compatible with the quantum circuit formalism. For a fair comparison, we benchmark our QKAN against Multi-Layer Perceptrons (MLPs) and KANs with comparable depth or parameter count.

### 4.2 Regression Tasks

We evaluate our method on synthetic regression tasks involving special functions known for their nontrivial structures and varying degrees of smoothness. Specifically, we fit Bessel functions of the first kind ($J_v$), second kind ($Y_v$), and modified Bessel functions ($I_v$, $K_v$). These functions pose a challenging benchmark due to their oscillatory, non-polynomial nature and variations in smoothness (Dutka, 1995). Each task uses 1000 training samples and 100 test samples.

Figure 1 reveals clear qualitative differences in the training dynamics of the two-layer models. QKAN achieves rapid and nearly monotonic loss reduction, with an early-stage exponential decay indicative of a well-conditioned optimization landscape. This stability suggests that the quantum-inspired activation functions introduce smooth, globally coherent nonlinear transformations that provide strong gradient signals. In contrast, while MLP shows steady improvement, it converges to a higher plateau. Classical KANs perform noticeably worse: their reliance on fixed basis expansions limits their ability to represent the rapid oscillations and varying curvature present in Bessel functions unless significantly more parameters are allocated.

The quantitative results in Table 1 further highlight QKAN's advantages. QKAN consistently achieves the lowest mean MSE on $J_v$ and $I_v$, and remains competitive on $Y_v$ and $K_v$, despite using fewer parameters than the 3-layer KAN and nearly matching the parameter count of the 6-layer MLP. Importantly, QKAN consistently and significantly outperforms both 2-layer and 3-layer classical KANs, despite the 3-layer KAN having more parameters. These observations suggest that QKAN's quantum-inspired, learnable activation functions provide a more expressive mechanism for modeling complex oscillatory behaviors than classical KANs with fixed basis functions.

### 4.3 Classification Tasks

We further assess the classification capabilities of QKAN on synthetic datasets commonly used to evaluate models with nonlinear decision boundaries: Gaussian blobs, concentric circles, and interleaving moons. These datasets evaluate a model's capacity to learn diverse geometric manifolds. For each dataset, we generate 1000 training samples and 100 test samples.

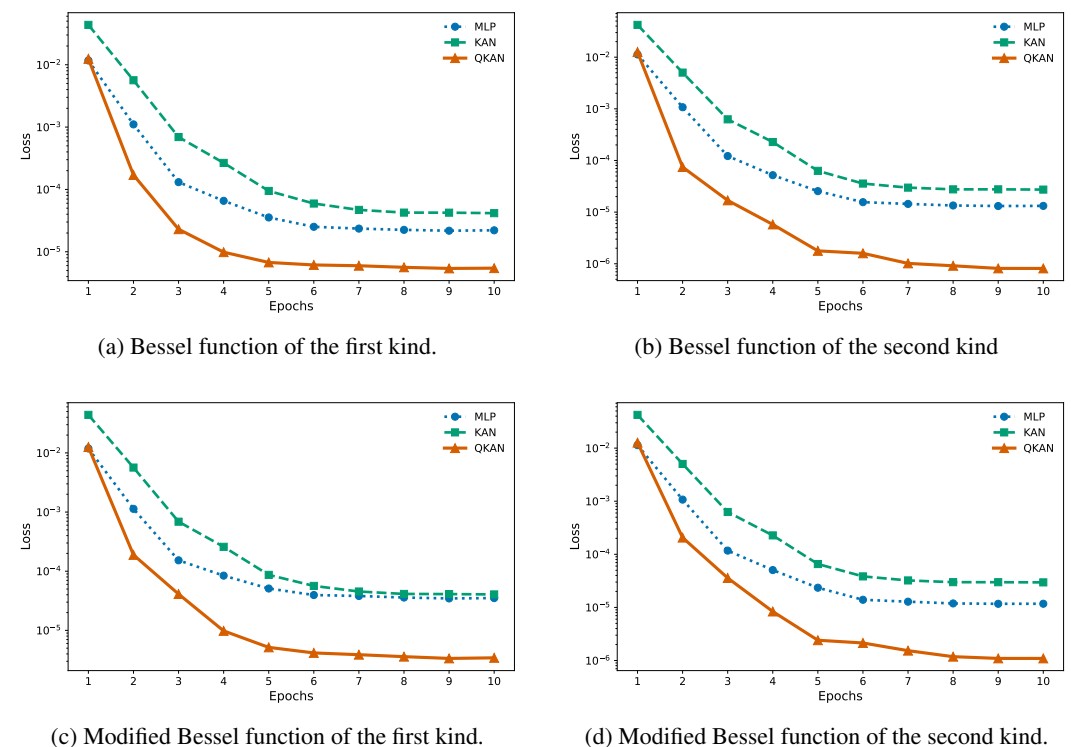

(a) Bessel function of the first kind.

(b) Bessel function of the second kind

(c) Modified Bessel function of the first kind.

(d) Modified Bessel function of the second kind.

Figure 1: Training loss trends for different Bessel function regression tasks.

| Method | # Params | $J_v$ | $Y_v$ | $I_v$ | $K_v$ |
|---|---|---|---|---|---|
| MLP (2 layers) | 21 | $11.42 \pm 9.93$ | $8.02 \pm 10.39$ | $18.68 \pm 13.99$ | $8.31 \pm 10.52$ |
| MLP (6 layers) | 141 | $5.81 \pm 0.97$ | $\mathbf{1.03} \pm 1.45$ | $22.48 \pm 10.62$ | $\mathbf{1.83} \pm 2.73$ |
| KAN (2 layers) | 60 | $19.86 \pm 11.96$ | $16.87 \pm 11.87$ | $21.85 \pm 12.07$ | $18.26 \pm 12.28$ |
| KAN (3 layers) | 160 | $13.25 \pm 14.85$ | $7.18 \pm 5.41$ | $12.25 \pm 15.87$ | $9.91 \pm 11.96$ |
| QKAN | 135 | $\mathbf{3.72} \pm 1.10$ | $1.11 \pm 1.09$ | $\mathbf{2.58} \pm 2.44$ | $1.88 \pm 2.37$ |

Table 1: Mean MSE ($\times 10^{-6}$) with standard deviation across 10 runs.

| Method | # Params | Gaussian | Circles | Moons |
|---|---|---|---|---|
| MLP (2 layers) | 21 | $68.40 \pm 16.90$ | $67.60 \pm 18.10$ | $81.50 \pm 11.30$ |
| MLP (6 layers) | 141 | $50.00 \pm 0.00$ | $50.90 \pm 2.85$ | $60.20 \pm 16.43$ |
| KAN (2 layers) | 60 | $87.30 \pm 6.63$ | $94.90 \pm 7.31$ | $87.30 \pm 2.11$ |
| KAN (3 layers) | 160 | $90.20 \pm 6.70$ | $97.60 \pm 3.84$ | $87.30 \pm 2.21$ |
| QKAN | 135 | $\mathbf{91.20} \pm 6.25$ | $\mathbf{99.60} \pm 0.52$ | $\mathbf{87.50} \pm 2.22$ |

Table 2: Mean classification accuracy (%) with standard deviation across 10 runs.

The classification results are presented in Table 2. QKAN achieves the best overall accuracy on all three datasets, with its most significant advantage on the challenging concentric circles task, where it attains near-perfect accuracy and low variance. In contrast, MLPs exhibit divergent performance: a shallow 2-layer network provides reasonable results, while a deeper 6-layer MLP frequently collapses to near-chance accuracy. This degradation highlights severe optimization issues—likely due to vanishing gradients or poorly conditioned updates—as deeper MLPs struggle to compose numerous local nonlinearities to model global patterns. Classical KANs perform adequately but show signs of saturation, failing to outperform QKAN even with more parameters in the 3-layer configuration.

Consequently, QKAN demonstrates a consistent ability to learn complex decision boundaries with a relatively modest parameter budget.

## 5 DISCUSSION

### 5.1 RESOURCE ANALYSIS

Consider a QKAN model with $L$ layers, where the $l$-th layer has width $N_l$, and the input dimension is $N_0$. Let $d$ denote the polynomial degree used in the GQSP transformations. Suppose we choose multi-layer in one ciucuit.

**Circuit width.** The circuit width scales logarithmically with the width of each layer. For layer $l = 0, \ldots, L-1$, the total number of qubits required up to layer $l+1$ satisfies

$$W(l+1) = W(l) + \mathcal{O}(\log_2 N_l), \tag{32}$$

with $W(0) = \mathcal{O}(1)$ accounting for the initial scalar encoding. Summing over all layers, the final circuit width is

$$W(L) = \mathcal{O}\left(\sum_{l=0}^{L-1} \log_2 N_l\right). \tag{33}$$

**Circuit depth.** The circuit depth grows more aggressively due to recursive polynomial transformation. The depth at layer $l+1$ satisfies the recurrence

$$D(l+1) = 2\left[N_l\left(d \cdot D(l) + \mathcal{O}(d)\right) + \mathcal{O}(\log_2 N_l)\right] + \mathcal{O}(1), \tag{34}$$

where $D(0) = \mathcal{O}(1)$. Unfolding this recurrence yields

$$D(L) = \mathcal{O}\left(d^L \cdot \prod_{l=0}^{L-1} N_l\right). \tag{35}$$

**Gate count.** The total number of quantum gates also follows a similar recurrence

$$G(l+1) = 2\left[N_l\left(d \cdot G(l) + \mathcal{O}(d)\right) + \mathcal{O}(\log_2 N_l)\right] + \mathcal{O}(1), \tag{36}$$

with $G(0) = \mathcal{O}(1)$. Therefore, the overall gate complexity at the final layer is

$$G(L) = \mathcal{O}\left(d^L \cdot \prod_{l=0}^{L-1} N_l\right). \tag{37}$$

Direct implementation of a deep QKAN faces a fundamental challenge: quantum circuit depth often incurs exponential complexity. To circumvent this barrier, we propose a modular strategy where the network is partitioned into its constituent layers, each implemented as a separate, shallow quantum circuit. These sub-circuits are executed in sequence, interleaved with mid-circuit measurements and classical feedforward of their results. This decomposition transforms the scaling of the problem from exponential with depth to linear with the number of layers. It fully preserves the parameterized structure and training dynamics of the original deep circuit while remaining practically feasible, thanks to the QKAN architecture's native compatibility with such split-and-merge processes.

### 5.2 FUNCTION APPROXIMATION PROPERTIES

Each layer of the QKAN applies a structured transformation to its input vector. Specifically, for the $l$-th layer, the output is given by

$$\Phi_l(\mathbf{x}) := \left[\sum_{j=1}^{N_{l-1}} q_{l,1,j}(x_j),\ \sum_{j=1}^{N_{l-1}} q_{l,2,j}(x_j),\ \cdots,\ \sum_{j=1}^{N_{l-1}} q_{l,N_l,j}(x_j)\right]^\top, \tag{38}$$

where $\mathbf{x} \in \mathbb{R}^{N_{l-1}}$ is the input to layer $l$, and each $q_{l,i,j}(x)$ denotes a trainable polynomial transformation from input dimension $j$ to output dimension $i$, for $i = 1, \ldots, N_l$. This transformation reflects

a layer-wise aggregation of independently modulated input features, consistent with the KAN layer construction but implemented via GQSP.

Because GQSP approximates polynomials $P(e^{ix})$ constrained by $|P(e^{ix})| \leq 1$ for all $x \in \mathbb{R}$, each function $q_{l,i,j}(x)$ lies in the set $\mathcal{S}$ defined by

$$\mathcal{S} := \left\{ \frac{1}{2} \Re \left( P(e^{i \arccos x}) + P(e^{-i \arccos x}) \right) \, \middle| \, |P(e^{ix})| \leq 1, \, \forall x \in \mathbb{R} \right\}, \tag{39}$$

which constitutes a restricted subset of real polynomials.

As depth increases, the recursive composition of bounded polynomial transformations leads to increasingly constrained output ranges. In particular, deeper QKANs tend to produce more tightly bounded functions due to the multiplicative effect of bounded approximations across layers. Consequently, more aggressive scaling of the target function during preprocessing is necessary to ensure it lies within the representable range of the network.

### 5.3 Potential Advantages

Classical KANs generate learnable activation functions through linear combinations of basis functions, requiring coefficient parameters that typically need regularization to prevent overfitting. In contrast, QKANs implement these functions using GQSP, which employs angle parameterization rather than explicit coefficients. This inherent parameterization eliminates the need for additional regularization, as the trigonometric structure of quantum operations naturally constrains the function space, potentially leading to more stable training and better generalization. Moreover, the modular structure of QKANs allows seamless integration with other quantum subroutines, potentially enabling end-to-end quantum machine learning pipelines that combine function approximation with quantum linear algebra and quantum optimization techniques.

## 6 Conclusion

In this work, we introduced QKANs, a novel quantum machine learning architecture that implements the mathematical structure of classical KANs using parameterized quantum circuits. QKANs leverage quantum computation to implement learnable activation functions without relying on predefined basis functions. By leveraging the GQSP framework, we constructed a quantum analogue of the KAN layer, which transforms input features through learned polynomial approximations encoded directly in quantum operations. Empirical evaluations on function approximation tasks demonstrate that QKANs accurately model complex nonlinear relationships and establish a new benchmark for expressive power in quantum machine learning.

Looking forward, several promising research directions emerge. The development of hybrid quantum-classical decomposition strategies could mitigate the exponential depth scaling observed in deep QKAN architectures. Further exploration of QKANs for scientific machine learning applications—such as solving differential equations, quantum chemistry simulations, and learning from quantum data—represents a natural extension. Additionally, rigorous theoretical analysis of QKAN expressivity and trainability would solidify their foundations and help identify clear pathways to quantum advantage. Investigating the impact of noise on QKAN performance will also be crucial for practical implementations on near-term quantum hardware.

As a fusion of mathematically grounded classical architecture with quantum computation, QKANs provide a promising paradigm that bridges fundamental approximation theory with emerging quantum computational capabilities. This work establishes a new foundation for expressive quantum machine learning and opens exciting avenues for future research at the intersection of quantum computing and mathematical deep learning.

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
