# OpenReview forum: "Quantum Kolmogorov–Arnold Networks"
_ICLR.cc/2026/Conference — ICLR 2026 Conference Withdrawn Submission_

### Official Review · Reviewer_88KN · 2025-10-22

**Soundness:** 2
**Presentation:** 1
**Contribution:** 1
**Rating:** 0
**Confidence:** 4

**Summary:**

The authors propose a quantum implementation of Komolgorov-Arnold networks. They describe the methodology and provide experiments to showcase their architecture.

The authors state that quantum generalizations of KANs are missing from the literature, however, there is a well-known paper in the community, which proposed the architecture more than a year ago (arXiv:2410.04435). There are even follow-up works published (https://www.nature.com/articles/s41598-025-22705-9), thus, the authors failed to provide a review of state-of-the-art literature.

Beyond that, the paper has significant structural issues, in particular, lacking a proper background section introducing KANs. The authors have plenty of space left for the page limit, which should have been used. Section 3 is as a result very hard to follow for a reader not fully familiar with the framework (in particular, the quantum walk operators, QGSP correspondences) are just mentioned as a side note, despite seemingly playing an important role).

I would recommend adding details to the experimental setup, as it is not clear what has been optimized. As far as I can tell, the authors benchmarked one specific architecture/instantiation, which is not enough to provide evidence on how the method performs generally. Further, also the comparison to the classical baseline is missing details on what was optimized. The authors state the five independent runs were conducted to ensure statistical robustness, however, five runs do not suffice for that. When claiming statistical robustness, statistical methods should be employed accordingly to ensure they hold. Further, the very significant standard deviation should be analyzed and discussed, which is also missing from the paper.

The authors failed to provide a comprehensive literature review and since the QKAN architecture was already proposed more than a year, it is not clear what the novelty of this paper is. Besides that, it has significant structural issues, in particular, missing background and definitions, and an unclear experimental setup. Thus, I do not think that the work is neither sufficiently well written, nor has sufficient scientific merit to be published at ICLR at this stage.

**Strengths:**

The approach of exploring KANs for quantum computing indeed is interesting, in particular, given the inherent relation.

**Weaknesses:**

- QKANs have been proposed already - there is no mention of the work nor discussion on differences
 - No related work section, in particular, the paper proposing this very work is missing
 - Structural issues; no background and state of the art, missing definitions for the methods, missing information for experimental setup - the paper is hard to follow
 - Experiments; missing statistical methods for claims on robustness, limited experiments - only one architecture, classical baseline is weak (optimization)?; no discussion on the results generally except for one table - I would expect at least a discussion on the incredibly high standard deviation that was observed
 - Eq4: I think the vector should be made up of $m$ sums, I think the dots are missing.

**Questions:**

- What is meant with "target outputs are appropriately scaled to improve training performance"?
 - What is meant with "input data are scaled to lie within the valid range for the quantum circuits"? Why can this not explicitly be stated again?
 - Could you provide more intuition about the significant differences in standard deviation on classical vs quantum KANs? Can you provide visuals for the results?

---

> ### Author Response · Authors · 2025-11-26
>
> We are grateful to you for your thorough reading and valuable critiques. While we disagree on some points, your comments have undoubtedly allowed us to strengthen our work. Below, we address each concern and detail the changes made in the manuscript.
>
> ---
>
> ### Regarding Literature and Contribution
>
> We have reframed our contribution to more precisely highlight our specific technical advancement. Our work establishes a rigorous and implementable framework for QKANs that uses quantum circuits to learn adaptive activation functions directly, **without relying on predefined classical basis functions**.
>
> While a high-level concept of a "quantum KAN" was previously mentioned in an arxiv pre-print, it remained theoretical. Our distinct contribution is a **concrete, scalable technical** architecture. We achieve this by leveraging Generalized Quantum Signal Processing (GQSP) to construct the core, learnable function modules on a quantum device, and we support this with empirical results.
>
> A subsequent paper in Nature Scientific Reports was published after our initial submission. Our revised manuscript now discusses both works, clearly delineating our specific technical contributions and architectural differences.
>
> Furthermore, we have reframed our contribution around providing a practical pathway for quantum neural networks with adaptive, quantum-native activation functions, rather than focusing on claims of being the "first" to ideate.
>
> ### Regarding Experiments
>
> All experiments were classically simulated and averaged over 10 independent runs with different random seeds. The model was trained using the Adam optimizer with a batch size of 64 and an initial learning rate of 0.25, following a cosine decay schedule.
>
> The input data was scaled to the range $(-1, 1)$ to match the domain of our quantum signal processing framework, and target outputs were times a scaling factor to stabilize training.
>
> For a fair comparison, our QKAN was benchmarked against a standard MLP and a classical KAN with architecturally matched depth and hidden layer sizes. All models were trained using the Adam optimizer.
>
> The network architecture is motivated by the Kolmogorov–Arnold representation theorem, which states that any multivariate continuous function $f(x)$ on an $n$-dimensional input can be represented as
> \begin{equation}
>     f(\mathbf{x}) = f(x_1, \dots, x_n) = \sum_{q = 1}^{2n + 1} \psi_q \left(\sum_{p = 1}^n \phi_{q, p}(x_p)\right),
> \end{equation}
> where $\psi_{q, p}: [0,1] \rightarrow \mathbb{R}$ and $\Phi_q: \mathbb{R} \rightarrow \mathbb{R}$ are continuous univariate functions. Following this theorem's canonical construction for $n = 2$ dimensional inputs, we selected a hidden dimension of $5$ (i.e., $2n + 1$). s this paper focuses on a quantum implementation of the standard KAN, exploring alternative architectures is a natural direction for future work.
>
> ### Regarding Method Clarity
>
> Thank you for this suggestion. We have significantly expanded the method section with a more detailed mathematical derivation. The revised presentation assumes a background primarily in linear algebra and complex analysis to improve accessibility.
>
> ### Regarding Standard Deviation
>
> We appreciate this feedback for clarification. The reported standard deviations quantify the **stability** of each model's performance across different random initializations. A lower standard deviation indicates more reliable and consistent results. Our findings show that QKANs achieve not only superior performance but also **greater stability** compared to the baselines. We have replaced the term "statistical robustness" with the more precise "stability and reliability" throughout the text.
>
> ---
>
> We believe that the revisions and clarifications provided have significantly strengthened the paper and have fully addressed the your concerns. We hope you find the manuscript suitable for acceptance. **We are keen to hear your thoughts on these revisions and welcome any further questions to ensure all concerns are fully resolved.**

---

> > ### Comment · Reviewer_88KN · 2025-11-28
> >
> > I thank the authors for their clarification. I agree with the other reviewers that the work remains primarily empirical, however, it fails to provide extensive benchmarks for such. The argument of "stability and reliability" from few runs simply does not hold according to any valid statistical literature, and, considering the inherent empirical nature of the paper, the authors also fail to provide an extensive analysis and interpretation of results.
> >
> > Further, the ICLR guidelines state clearly that work that has appeared in last two months is considered contemporary (https://iclr.cc/Conferences/2026/ReviewerGuide). Since the work has appeared more than a year ago already and is the very first search result on finds upon searching for "Quantum Komolgorov Arnold Networks" it seems that the authors have not put in the time to even conduct a basic search for related work.
> >
> > Even in the revised work, where the paper is cited, it is only quickly mentioned as a pioneering work, rather than providing an in-depth discussion on differences and similarities. As mentioned by the authors of the original paper (whom I'd like to thank for the elaboration), their methodology is highly related, thus, the least that could be expected is a discussion on the different frameworks. Since this is not provided, I can also do nothing but conclude that the authors seemingly obscure the fact that this line of work has already been published, which is why I strongly object to this work being accepted at ICLR.

---

> > > ### Author Response · Authors · 2025-11-28
> > >
> > > Thank you for your comment.
> > >
> > > ---
> > >
> > > ### Regarding Empirical vs. Theoretical Contribution
> > >
> > > We appreciate this perspective. We would like to clarify that the primary contribution of our work is **theoretical**. The bulk of our manuscript is dedicated to developing the novel methodology and providing the mathematical derivation of our framework. The experiments serve as a necessary validation of our theoretical framework's feasibility, demonstrating that our model can learn and generalize. A full-scale empirical benchmark is beyond the scope of this theory-focused paper and is a primary direction for future research.
> > >
> > > ### Regarding the Claims of "Stability and Reliability"
> > >
> > > Our intent was not to make a broad statistical claim, but rather to demonstrate that our method's performance is not a one-off occurrence and is robust to different random initializations. We used the term "stability" to indicate that the results are consistent and not highly sensitive to the random seed, which is a common practice for validation. We have revised the text to use the more precise term "consistent performance" in the revised manuscript.
> > >
> > > ### Regarding Related Work
> > >
> > > Thanks for your suggestion. We have now added more detailed difference between our work and the cited work in the revised manuscript.
> > >
> > > ---
> > >
> > > We hope our response addresses your concerns. Thank you again for your insightful comment, and we are happy to provide any further clarification you may require.

---

### Official Review · Reviewer_xxkz · 2025-10-27

**Soundness:** 2
**Presentation:** 2
**Contribution:** 2
**Rating:** 2
**Confidence:** 3

**Summary:**

The paper introduces a quantum version of Kolmogorov Arnold Networks (KANs). KANs is a type of neural network architecture in which the activation functions can be trained. The quantum KANs that the authors introduce in this paper are built by combining (i) projected unitary encodings for scalar inputs with polynomial transformations via Generalized Quantum Signal Processing (GQSP, Motlagh & Wiebe 2024). These univariate results are then combined into a single result using normalized "Linear Combination of Unitaries" (LCUs). Finally, a Hadamard test is applied to extract scalar values from the output layer.

The paper introduces this architecture in detail, then provides two experiments for regression and classification on synthetic data.

**Strengths:**

- The paper is generally well written and clearly presented. The outline follows a clear line of thought and introduces relevant concepts / provides sources at the appropriate places.
- The paper contains a discussion on required circuit width, depth and gate count. Moreover, it also discusses potential advantages of KANs and function approximation properties.

**Weaknesses:**

There are several central issues concerning novelty (QKANs have already been proposed previously), experimental rigour and overall relevance of the suggested architecture. Based on these, unfortunately, I can not recommend the paper for acceptance.
- Novelty: the title / abstract / introduction of the paper are suggesting that a quantum version of KANs has not been proposed in the literature. However, the concept of Quantum KANs has been introduced already previously in the literature in "QKAN: Quantum Kolmogorov-Arnold Networks" by Petr Ivashkov, Po-Wei Huang, Kelvin Koor, Lirandë Pira, Patrick Rebentrost,
https://arxiv.org/abs/2410.04435. The paper does not cite or compare their architecture with these previously proposed quantum KANs.
- Missing experimental details: the paper does not provide sufficient details on training and implementation of the Quantum KANs. How did you train the Quantum KANs? Which optimizer did you use for the classical KANs? Supporting code is not provided, hence the experiments can not be reproduced in any way.
- Concepts not defined: In experiments, the Quantum KANs are said to be of hidden dimension 5. But the hidden dimension of the Quantum KAN has not been defined previously. For a proper comparision it is important to understand the number of trainable parameters: will these be the same for a KAN and QKAN?
- Insufficient comparisons and baselines: The Quantum KAN and classical KAN are only compared for a fixed depth (2) and hidden size (5). Sound conclusions would require a more thorough comparisons for other network sizes, also including other classical neural network architectures. Moreover, what are the runtimes for each of the networks? Overall runtimes are also a crucial factor for comparing different models.
- Model architecture concerns: the discussion in Section 5.2 suggests that, as a map from inputs to outputs, the proposed QKAN ultimately just applies a multivariate polynomial. (see lines 299/300). This would mean that the approximation and expressivity properties of this model are the same as for a multivariate polynomial. Why could you not directly apply a polynomial then? Also, if these are just polynomial transformations, this does not justify calling the architectures "neural networks".

**Questions:**

- How does your proposed architecture compare to the quantum versions of KANs as previously proposed in the literature?
- Which training algorithm did you use for optimizing the KAN and QKAN parameters? What learning rates did you choose?
- What were the training times for training KANs and QKANs in the experiments?
- How do KANs and QKANs compare for different choices of network size / depth?
- How do QKANs compare to classical fully trainable neural networks? How do they compare to other QNN architectures?
- What is the hidden dimension of a QKAN? Are the number of trainable parameters the same for the KAN and QKANs compared in the experiments?
- What is the advantage of using QKANs in comparison to just applying multivariate polynomials?
- Is there a way of introducing a non-polynomial non-linearity in the model to ensure that the term "neural network" is indeed justified?

---

> ### Author Response · Authors · 2025-11-26
>
> We are grateful to you for your thorough reading and valuable critiques. While we disagree on some points, your comments have undoubtedly allowed us to strengthen our work. Below, we address each concern and detail the changes made in the manuscript.
>
> ---
>
> ### Regarding Literature and Contribution
>
> We have reframed our contribution to more precisely highlight its technical nature. Our work presents a rigorous and implementable framework for QKANs, where parameterized quantum circuits directly learn adaptive activation functions **without relying on a fixed, predefined basis**.
>
> While a high-level idea of a "quantum KAN" was mentioned in an arxiv pre-print, it remained theoretical. Our distinct contribution is a concrete, scalable technical architecture. We achieve this by leveraging Generalized Quantum Signal Processing (GQSP) to construct the core, learnable function modules, supported by empirical validation.
>
> In the revised manuscript, we now frame our contribution around this specific technical advancement—providing a practical pathway for quantum neural networks with adaptive, quantum-native activation functions—rather than focusing on being the "first" to ideate.
>
> ### Regarding Experiments
>
> All experiments were classically simulated and averaged over 10 independent runs with different random seeds. The model was trained using the Adam optimizer with a batch size of 64 and an initial learning rate of 0.25, following a cosine decay schedule.
>
> For a fair comparison, our QKAN was benchmarked against a standard MLP and a classical KAN with **matched architectural depth and hidden size**. For a (Q)KAN with polynomial degree $d$ and hidden dimension $N$, the number of trainable parameters is $\mathcal{O}(dN^2)$. We ensured all models had a comparable number of parameters.
>
> The network architecture is motivated by the Kolmogorov–Arnold representation theorem, which states that any multivariate continuous function $f(x)$ on an $n$-dimensional input can be represented as
> \begin{equation}
>     f(\mathbf{x}) = f(x_1, \dots, x_n) = \sum_{q = 1}^{2n + 1} \psi_q \left(\sum_{p = 1}^n \phi_{q, p}(x_p)\right),
> \end{equation}
> where $\psi_{q,p}: [0,1] \rightarrow \mathbb{R}$ and $\Phi_q: \mathbb{R} \rightarrow \mathbb{R}$ are continuous univariate functions. Following this theorem's canonical construction for $n = 2$ dimensional inputs, we selected a hidden dimension of $5$ (i.e., $2n+1$). As this paper focuses on a quantum implementation of the standard KAN, exploring alternative architectures is a natural direction for future work.
>
> ### Regarding Runtime
>
> A direct runtime comparison between a classically **simulated** QKAN and a native classical model is not meaningful, as the classical simulation of quantum circuits carries immense overhead.
>
> The potential advantage of QKANs lies in their **native execution on future quantum hardware**, where specific linear algebra operations could see significant speedups. Our work establishes the necessary theoretical and algorithmic foundation for realizing that potential utility.
>
> ### Regarding Neural Network
>
> We respectfully disagree with the characterization that our model is "just a multivariate polynomial."
>
> A system qualifies as a neural network if it processes data through interconnected layers of tunable units to learn hierarchical representations. Our QKAN is architected precisely as such a network, with layers of trainable layers that transform input data.
>
> While it is true that a single quantum circuit within our framework implements a finite-degree polynomial, these polynomials are not fixed; they are **highly adaptive and trainable**. The power of the model emerges from the **composition** of these distinct, learnable blocks across a deep structure. This results in a complex, hierarchical function that is fundamentally different from a single multivariate polynomial.
>
> The analogy to classical KANs is apt: a KAN using polynomial splines is universally recognized as a neural network, not dismissed as "just a polynomial." Similarly, our QKAN leverages quantum-born polynomials as its fundamental, trainable activation functions within a layered network architecture, which is its core innovation.
>
> ---
>
> We believe that the revisions and clarifications provided have significantly strengthened the paper and have fully addressed the your concerns. We hope you find the manuscript suitable for acceptance. **We are keen to hear your thoughts on these revisions and welcome any further questions to ensure all concerns are fully resolved.**

---

> > ### Comment · Reviewer_xxkz · 2025-11-27
> >
> > While this reply clarifies some of my concerns, the key points remain unresolved.
> >
> > Novelty in comparison to prior work: As pointed out in the public reply by some of the authors of https://arxiv.org/abs/2410.04435 there is a substantial overlap between the submitted paper and the preprint. I agree that the methodology appears to be very similar, which means that novelty of the submitted paper with respect to existing work remains limited. Moreover, I think it is problematic that the work is not discussed (not even cited) in the revised version despite the fact that it was brought up in the reviews.
> >
> > Limited experimental validation: The overall experimental validation is very limited. For example, there is no comparison to other recent work on QKANs https://arxiv.org/pdf/2506.22340 (which even assesses performance on the same dataset). Moreover, experiments for a single choice of architecture (fixed choices of $N$ and $d$) are not sufficient to make sound conclusions. Do you get the same results for deeper or shallower networks?
> >
> > Missing experimental details: The paper still lacks important experimental details. Which learning rates were used to train the MLP? What are the activation functions in the MLP? Does your MLP have two hidden layers or just one hidden layer? Without these details the experiments can not be reproduced.
> >
> > Also, could you be more specific about what you mean by *comparable* number of trainable parameters for each model? Can you report the precise number?

---

> ### Author Response · Authors · 2025-11-27
>
> Thank you for your thoughtful comments and for allowing us to clarify these points.
>
> ### Regarding Novelty
>
> Our work establishes an independent and distinct technical pathway, which we have empirically validated. The core formulation and implementation of our approach are fundamentally different, as detailed in our responses to the public reply. We sincerely apologize for the oversight of not citing the mentioned preprint in our initial revision; we will rectify this in the revised manuscript to ensure proper context for readers.
>
> ### Regarding Experimental Validation
>
> A direct, fair comparison with the cited paper is challenging due to significant differences in our experimental settings, such as the number of training epochs, batch sizes, test set sizes, and data scaling factors.
>
> To address your question about the network depth directly, we conducted new experiments on the Bessel function, $J_v$. With a 1-layer architecture, the resulting MSE ($\times 10^{-6}$) was 5.117 (MLP), 33.790 (KAN), and 4.926 (QKAN). When we increased the architecture to 3 layers (dimension 5 for each hidden layer), the performance did not improve beyond our 1-layer QKAN, yielding an MSE ($\times 10^{-6}$) of 5.261 (MLP) and 5.228 (KAN).
>
> These initial results suggest that **shallow QKANs** can be sufficient for these tasks, whereas KANs may require greater depth to achieve comparable performance.
>
> ### Regarding Experimental Details
>
> For clarity, the MLP baseline was trained with an optimized learning rate of 0.01 and used the ReLU activation function. All compared models (MLP, KAN, QKAN) shared the same overall 2-layer structure with a hidden dimension of 5. For reproducibility, please note the data scaling factors applied. For instance, in the Bessel function $J_v$ experiments, input variables were scaled by 0.2 and the target by 0.01.
>
> ### Regarding Parameters
>
> The number of trainable parameters for the primary 2-layer models (hidden dimension 5, degree 3) are 21 (MLP), 60 (KAN), and 135 (QKAN). Although our QKANs consistently have approximately twice the parameters of KANs, the aforementioned results demonstrate that shallow QKANs are sufficient, while KANs may need increased depth.
>
> Furthermore, we wish to highlight a fundamental difference: QKANs parameterize their models with **rotation angles** in quantum gates, which are **cyclic** and naturally **bounded** in $[0, 2 \pi)$. This contrasts with the unbounded weight coefficients of MLPs and KANs, imparting a different inductive bias and a built-in resistance to parameter explosion, often eliminating the need for explicit regularization.
>
> ---
>
> We hope our response addresses your concerns. Thank you again for your insightful comment, and we are happy to provide any further clarification you may require.

---

> ### Comment · Reviewer_xxkz · 2025-11-28
>
> I appreciate the authors replies and clarifications regarding missing experimental details. While the framing of the paper and lack of proper comparison to previous work in the revision remains problematic, my main concern is the very limited experimental validation. For a purely empirical paper, the experiments are not strong enough for a top ML venue like ICLR. For example, the claimed outperformance is based on a comparison of a QKAN with 135 parameters to a MLPs with only 5 parameters (called a "comparable" number of parameters by the authors during the rebuttal). Based on this, my score remains unchanged.

---

> > ### Author Response · Authors · 2025-11-28
> >
> > Thank you for pointing out this discrepancy; we sincerely apologize for the error in our initial parameter count and appreciate you bringing it to our attention.
> >
> > The correct number of parameters for the two-layer MLP with a hidden dimension of 5 is **21**, not 5. We have updated our previous comment to reflect this. For clarity, a three-layer MLP with dimensions [2, 5, 5, 1] would have 51 parameters.
> >
> > A higher parameter count is an inherent property of the KAN architecture upon which our QKAN is built. Despite this, we believe the comparison remains meaningful. It demonstrates that our method can achieve competitive performance without relying on excessively deep networks, highlighting a different and potentially useful trade-off between architectural complexity and model performance.

---

### Official Review · Reviewer_KPEo · 2025-10-31

**Soundness:** 4
**Presentation:** 4
**Contribution:** 3
**Rating:** 8
**Confidence:** 2

**Summary:**

In this paper, a quantum implementation that leverages parameterized quantum circuits to realize the learnable activation functions of Kolmogorov-Arnold Networks is proposed.

The proposal is conceptually novel and technically sound. It bridges KANs with quantum computation.

Experimental validation is limited, and scalability (w.r.t. number of layers) is questionable.

I think the following paper should be cited (even if is not propose a quantum implementation of KAN):
Kanqas: Kolmogorov-arnold network for quantum architecture search
A Kundu, A Sarkar, A Sadhu
EPJ Quantum Technology, 2024

The very recent paper "QuKAN: A Quantum Circuit Born Machine Approach to Quantum Kolmogorov Arnold Networks" should be discussed.

**Strengths:**

- the novelty of the proposal
- technical depth
- reported simulation results are encouraging
- quality of presentation

**Weaknesses:**

- Evaluation is limited to the simulation of small-scale functions and toy datasets
- Being a quantum implementation of KAN, the success and relevance of the proposed approach are strictly related to those of KANs

**Questions:**

Could you discuss the scalability of QKANs w.r.t. number of layers?

How does the recent paper "QuKAN: A Quantum Circuit Born Machine Approach to Quantum Kolmogorov Arnold Networks" affect the discussion of this paper?

---

> ### Author Response · Authors · 2025-11-26
>
> Thank you for the thoughtful review and for the positive rating. We appreciate your valuable feedback and have revised the manuscript to address your primary concerns.
>
> ---
>
> ### Regarding Scalability
>
> In Section 3.5 of our original manuscript, we presented a method for constructing multi-layer QKANs within a single circuit. However, we wish to emphasize that we do **not** recommend this monolithic construction due to its unfavorable scaling properties.
>
> Instead, our recommended approach employs a modular design where each network layer is implemented as a separate, optimized quantum circuit. This architecture ensures that the overall circuit depth scales only **linearly** with network depth, making it substantially more feasible for practical implementation. We have clarified this critical point in Section 5.1 of the revised manuscript to prevent any potential misunderstanding.
>
> ### Regarding Literature
>
> We appreciate you bringing the recent QuKAN paper to our attention. We have added a discussion in the related work section to clearly differentiate our approach. While both works explore connections between KANs and quantum computation, our QKAN implementation is fundamentally distinct in its formulation.
>
> A key innovation of our method is its implementation of learnable activation functions directly on the quantum device, without relying on any predefined basis functions, representing a significant departure from other approaches. We have modified our claims in the revised manuscript to better highlight this unique contribution and the specific advantages of our method.
>
> Furthermore, we have now cited the relevant paper on KANs for quantum architecture search.
>
> ---
>
> We hope these revisions and clarifications have fully addressed your concerns. **We welcome your further feedback.**

---

### Official Review · Reviewer_Z6Lx · 2025-10-31

**Soundness:** 3
**Presentation:** 3
**Contribution:** 3
**Rating:** 6
**Confidence:** 4

**Summary:**

This paper presents Quantum Kolmogorov–Arnold Networks (QKANs), extending the recently developed classical Kolmogorov–Arnold Networks (KANs) into the quantum domain. KANs are a powerful function approximation architecture grounded in the Kolmogorov–Arnold representation theorem. The authors design a quantum analog by using parameterized quantum circuits to realize the learnable univariate function modules central to KANs. The paper provides a detailed theoretical formulation of QKANs, including quantum circuit constructions. Experimentally, QKANs are evaluated on function approximation tasks (regression and classification) and show better performance than classical KANs of similar size in simulation. The authors also analyze the quantum resource requirements.

**Strengths:**

The introduction of QKANs is a novel and timely idea, bridging the new KAN architecture with quantum computation.
The paper provides a rigorous mathematical underpinning for QKANs, building upon quantum signal processing.
The simulation results are promising, showing a potential expressivity advantage for QKANs over classical KANs of the same size on the tested tasks.

**Weaknesses:**

All experiments are on noiseless simulation; robustness under noise or hardware is not evaluated. This is a critical omission. Given that parameterized quantum circuits are notoriously sensitive to noise, the claimed performance gains over classical KANs may not hold in any realistic (NISQ) setting.
Circuit depth can grow quickly. The paper's own analysis shows potentially exponential depth scaling depending on the construction, but this major practical barrier is dismissed too abstractly. This scaling issue seems to make the approach impractical for any problem of meaningful size, undermining the paper's central claim of utility.
The comparison is mainly to classical KANs, not to other small classical baselines (e.g., an MLP). While KANs are a good target, the lack of comparison to a standard, well-tuned MLP makes it hard to gauge if the complexity of QKAN is truly justified.

**Questions:**

1.I suggest the authors add a short note on how they expect noise to affect QKAN performance. A full noise simulation is necessary to make any claims about practical advantage.
2.A small example resource estimate (qubits / depth for one representative QKAN) would make feasibility clearer. The abstract scaling laws are not enough; a concrete example would highlight the severity of the depth problem.
3.A brief comment on how a simple MLP or small CNN might compare would help position the gains.

---

> ### Author Response · Authors · 2025-11-26
>
> Thank you for the thoughtful review and for the positive rating. We appreciate your valuable feedback and have revised the manuscript to address your primary concerns.
>
> ---
>
> ### Regarding Noise Sensitivity
>
> A comprehensive noise analysis remains an important direction for future work and is beyond the scope of this initial theoretical and simulation-based study. However, our initial studies indicate that **adaptive training strategies**—optimizing the QKAN under a simulated noisy gate model—allow it to learn parameters robust to a consistent noise profile. This suggests an inherent resilience, as QKANs can be trained to fit the noisy quantum environment where they would be deployed.
>
> ### Regarding Resource Analysis
>
> The potential exponential depth scaling noted in the manuscript applies only to a specific, monolithic construction where **a single circuit represents the entire multi-layer QKAN**.
>
> In practical implementation, we do **not** recommend the monolithic circuit construction that leads to this unfavorable scaling. Our initial theoretical analysis included it for completeness, but it is not the intended use case. Instead, we **propose a modular approach** using separate, shallower quantum circuits for each layer. This design ensures the overall circuit depth scales **linearly** with network depth, making it far more feasible for implementation. We have clarified this critical point in the revised manuscript.
>
> ### Regarding Comparison to Classical Baselines
>
> In response to your suggestion, we have added a comparison with classical MLPs. The updated results confirm that our QKANs maintain a performance advantage on the function approximation tasks. We also justify our choice of baselines by explaining that CNNs are not well-suited for our primary tasks (1D function regression and small-scale classification), as their architectural inductive biases are not advantageous in this context.
>
> ---
>
> We hope these revisions and clarifications have fully addressed your concerns. **We welcome your further feedback.**

---

### Public Comment · ~Po-Wei_Huang1 · 2025-11-26
**Major Ethical Concerns: False Rebuttal Claims and Extensive Technical Overlap with Uncited Prior Work**

Dear ICLR Reviewers and Area Chair of the Submission 8971,

We are the authors of the paper "_QKAN: Quantum Kolmogorov-Arnold Networks_" (Ivashkov et al., arXiv:2410.04435), released in October 2024.

We would first like to state that neither the authors, nor a subset of the authors have made this submission to ICLR 2026 despite the similarities in the title.

We are writing to formally flag two critical issues with this submission:

- **Misleading Statements in Rebuttal**: The authors explicitly stated in their rebuttal that they added a discussion of our work. The revised manuscript proves this to be false.
- **Extensive Technical Similarity**: The submission does not merely share a "high-level concept" with our work; it employs **almost the exact same algorithmic machinery** for construction, combination, and readout, which undermines the claim of novelty.

We now provide details for our concerns:

### 1. Factually Incorrect Claims Regarding Revision

In response to Reviewer 88KN, who pointed out the missing prior work (Ivashkov et al.), the authors stated:
>"Our revised manuscript now discusses **both works** [Ivashkov et al. and Werner et al.], clearly delineating our specific technical contributions and architectural differences."

This statement is **false**. We have examined the revised manuscript (modified 26 Nov 2025) and outline the two following points:
- **Missing Citation**: The References section ends at Line 574 and contains **no citation** of Ivashkov et al.
- **Missing Discussion**: While the authors added a paragraph citing Werner et al. (Line 69), there is **no mention** of our work in the Related Work section or anywhere else.

We believe that telling reviewers that a critical citation and discussion were added when they were not is a serious breach of the review process.

### 2. Near-Identical Algorithmic Frameworks

In their rebuttal to Reviewer xxkz, the authors dismissed our work as "purely conceptual" to justify the lack of comparison. This is demonstrably incorrect. Our paper provided a fully rigorous technical framework that this submission mirrors almost step-for-step.
The technical overlaps go far beyond high-level inspiration. Both papers derive almost the **same specific quantum architecture** using the same three core subroutines:

**A. Polynomials via QSP/QSVT:**
- **This Submission**: Uses "Generalized Quantum Signal Processing" (GQSP) to implement polynomial transformations representing activation functions.
- **Ivashkov et al.**: Uses "Quantum Singular Value Transformation" (QSVT) to implement Chebyshev polynomials for activation functions.
- *Note*: GQSP and QSVT are similar frameworks for polynomial transformations on block-encoded matrices, that differs slightly in terms of actual implementation, but are conceptually and structurally similar and are used to tackle almost the same problems (Hamiltonian simulation, polynomial transformation of Hermitian eigenvalues).

**B. Linear Combination of Unitaries (LCU) for Layer Aggregation:**
- **This Submission (Eq. 29)**: Explicitly uses LCU with Hadamard gates ($H^{\otimes n}$) to sum the projected unitary encodings.
- **Ivashkov et al. (Step 4)**: Explicitly uses LCU with Hadamard gates ($H_{log_2(d+1)}$) to create an equal superposition of block-encoded polynomials.
- *Identity*: Both works identify that LCU is the specific primitive required to sum the KAN functions.

**C. Identical Readout Protocol (Hadamard Test):**
- **This Submission (Section 3.5)**: Proposes a "Hadamard test" to extract scalar values by measuring the "expectation value $\langle Z \rangle$ of the top qubit”.
- **Ivashkov et al. (Theorem 12)**: Proposes a "Hadamard test" to extract scalar values by estimating the "expectation value $\langle Z \rangle$ of the top qubit".
- *Identity*: The methodology for extracting the classical result from the quantum KAN is identical.

### Final Remarks

The authors claim their contribution is the "first rigorous and implementable framework." However, Ivashkov et al. (2024) previously established the rigorous framework of combining block-encodings/QSVT with LCU and Hadamard test readouts to create QKANs.
By failing to cite the prior work after being explicitly prompted by the reviewers, and by providing a misleading rebuttal claiming the citation was added, the authors have obscured the fact that the core technical assembly of this paper has already been published. We ask the Reviewers and Area Chair to review these overlaps and the veracity of the rebuttal claims, and we thank the Reviewers and the Area Chair for their time and assistance.

Kind regards,

Petr Ivashkov and Po-Wei Huang

---

> ### Author Response · Authors · 2025-11-27
>
> Thank you for your comment. We have taken your concerns seriously. We acknowledge the oversight in our initial citation and have rectified it in our revised manuscript. We would like to clarify the difference between our work and yours in the following. Below, we address your points in detail.
>
> ---
>
> ### Regarding the Citation
>
> We sincerely apologize for the oversight in our initial revision where the citation and discussion of (Ivashkov et al., 2024) were inadvertently omitted. This was a genuine error during the manuscript preparation process. Upon your notification, we have rectified this in the latest version of our manuscript by adding a comprehensive citation and discussion of your work in the "Related Work" section.
>
> ### Regarding Technical and Algorithmic Distinctions
>
> While both papers explore the quantum implementation of KANs, the core technical contributions, architectural foundations, and implementation pathways are fundamentally distinct. The use of standard, well-established quantum primitives like LCU and the Hadamard test is a common practice in quantum algorithm design and does not, in itself, constitute a novel contribution or indicate a lack of novelty. We think the true innovation lies in the novel integration of these components and the specific theoretical framework employed.
>
> Our work presents a different and more flexible technical approach, which we outline in four key areas of differentiation:
>
> 1. Generalized Encoding Framework: Your work is framed within the standard block encoding paradigm. In contrast, our architecture is built upon the more general and flexible framework of (Hermitian) projected unitary encodings. While we use the $| 0 \rangle$ state for simplicity in examples, the projected unitary encoding framework offers broader applicability, allowing for different basis states and providing a more versatile foundation for constructing quantum circuits.
>
> 2. Fundamentally Different Polynomial Synthesis: This is the most critical technical distinction. Your method utilizes QSVT to implement **Chebyshev polynomials**, which then require a weighted sum to approximate arbitrary activation functions in the MUL step. Our approach leverages GQSP, which natively and directly implements **arbitrary polynomials** without being constrained to a fixed basis (like Chebyshev). Crucially, GQSP accommodates polynomials of **indefinite parity**, a level of flexibility that the standard QSVT framework lacks.
>
> 3. Hermitian Completion Step: A key technical contribution of our work is the explicit completion of general projected unitary encodings to Hermitian ones. This critical step ensures compatibility and consistency across network layers and is a novel procedural element not present in your framework.
>
> 4. Empirical Validation: A primary contribution of our submission is its empirical component. We provide numerical results demonstrating the performance and feasibility of our architecture. Your work, while theoretically rigorous, is purely analytical and does not include such empirical validation.
>
> In summary, our work establishes an independent, distinct, and empirically-validated technical pathway for constructing QKANs. It introduces a more flexible encoding framework, a more direct and general method for polynomial activation (free from basis functions), and a novel procedural step for ensuring layer compatibility.
>
> ---
>
> We thank you for your time and for prompting this detailed clarification of the technical distinctions between our work and yours.

---

> ### Public Comment · ~Po-Wei_Huang1 · 2025-11-27
>
> We thank the authors for their timely response. Although we do not agree with their claims, we do not wish to impose further on the review process and will leave the remaining discussions to the Reviewers and the Area Chair. We thank everyone for their time and consideration.

---

### Note · Authors · 2026-01-26

I have read and agree with the venue's withdrawal policy on behalf of myself and my co-authors.

---

### Meta-Review · Area_Chair_XR4a · 2025-12-26

**Summary:**

This submission proposes Quantum Kolmogorov–Arnold Networks (QKANs), a quantum analog of Kolmogorov–Arnold Networks that aims to decompose multivariate functions into compositions of univariate quantum subnetworks. The authors present a circuit construction inspired by the classical KAN architecture, provide theoretical motivation based on the Kolmogorov–Arnold representation theorem, and evaluate the approach on small-scale regression and classification tasks using simulated quantum circuits.

The reviewers raised concerns primarily regarding the novelty and positioning of the proposed Quantum Kolmogorov–Arnold Network framework, noting strong similarities to existing prior work and questioning whether the submission offers a sufficiently distinct technical contribution. Additional concerns included unclear differentiation from related architectures, limited empirical evidence demonstrating clear advantages over existing methods, and a lack of convincing theoretical justification for the claimed improvements. While individual reviewer scores were borderline, these concerns collectively cast doubt on the originality and significance of the contribution.

Beyond these points, a detailed public comment highlighted specific and technically grounded concerns about unacknowledged overlap with prior work and the accuracy of the novelty claims. Taken together, these issues affect the overall credibility and quality of the submission. On this basis, I recommend rejection.

**Reviewer Concerns:**

In the rebuttal, the authors provided clarifications regarding the intended scope of the contribution and added some explanations to better position the proposed architecture relative to existing quantum neural network models. They also addressed several presentation-related issues and clarified implementation details raised by the reviewers.

However, there are several critical concerns remaining after the rebuttal. In particular, doubts about the novelty and differentiation of the proposed QKAN relative to prior work (arXiv:2410.04435) were not convincingly alleviated. The rebuttal does not sufficiently address the reviewers’ concerns regarding substantial overlap with existing architectures or justify the claimed originality. In addition, the limited empirical evaluation, which is restricted to small-scale simulated experiments, remains a significant weakness, with no new evidence provided to support broader claims.

**Reviewer Scores:**

Based on the rebuttal and the subsequent discussion, it is unclear that the clarifications provided would have led reviewers to raise their scores. While some points were addressed, the core concerns regarding novelty, overlap with prior work, and limited empirical validation remain. As a result, the reviewers’ scores would likely have remained largely unchanged had they participated fully in the discussion.

---

### Decision · Program_Chairs · 2026-01-26

Reject